# *RTG* Signaling Sustains Mitochondrial Respiratory Capacity in *HOG1*-Dependent Osmoadaptation

**DOI:** 10.3390/microorganisms9091894

**Published:** 2021-09-06

**Authors:** Nicoletta Guaragnella, Gennaro Agrimi, Pasquale Scarcia, Clelia Suriano, Isabella Pisano, Antonella Bobba, Cristina Mazzoni, Luigi Palmieri, Sergio Giannattasio

**Affiliations:** 1Department of Biosciences, Biotechnologies and Biopharmaceutics, University of Bari “Aldo Moro”, 70125 Bari, Italy; gennaro.agrimi@uniba.it (G.A.); pasquale.scarcia@uniba.it (P.S.); cleliasuriano95@gmail.com (C.S.); isabella.pisano@uniba.it (I.P.); luigi.palmieri@uniba.it (L.P.); 2Institute of Biomembranes, Bioenergetics and Molecular Biotechnologies, National Research Council, 70125 Bari, Italy; a.bobba@ibiom.cnr.it (A.B.); s.giannattasio@ibiom.cnr.it (S.G.); 3Department of Biology and Biotechnology ‘Charles Darwin’, Pasteur Institute-Cenci Bolognetti Foundation, Sapienza University of Rome, Piazzale Aldo Moro 5, 00185 Rome, Italy; cristina.mazzoni@uniroma1.it

**Keywords:** *RTG* signaling, *HOG1*, osmoadaptation, mitochondria, respiratory capacity, stress response, metabolism

## Abstract

Mitochondrial *RTG*-dependent retrograde signaling, whose regulators have been characterized in *Saccharomyces cerevisiae*, plays a recognized role under various environmental stresses. Of special significance, the activity of the transcriptional complex Rtg1/3 has been shown to be modulated by Hog1, the master regulator of the high osmolarity glycerol pathway, in response to osmotic stress. The present work focuses on the role of *RTG* signaling in salt-induced osmotic stress and its interaction with *HOG1*. Wild-type and mutant cells, lacking *HOG1* and/or *RTG* genes, are compared with respect to cell growth features, retrograde signaling activation and mitochondrial function in the presence and in the absence of high osmostress. We show that *RTG2,* the main upstream regulator of the *RTG* pathway, contributes to osmoadaptation in an *HOG1*-dependent manner and that, with *RTG3*, it is notably involved in a late phase of growth. Our data demonstrate that impairment of *RTG* signaling causes a decrease in mitochondrial respiratory capacity exclusively under osmostress. Overall, these results suggest that *HOG1* and the *RTG* pathway may interact sequentially in the stress signaling cascade and that the *RTG* pathway may play a role in inter-organellar metabolic communication for osmoadaptation.

## 1. Introduction

Mitochondrial retrograde signaling is an evolutionarily conserved pro-survival pathway sustaining metabolic adaptation especially, but not exclusively, in the case of mitochondrial dysfunction. This pathway has been characterized in its molecular details in *Saccharomyces cerevisiae* yeast cells lacking mitochondrial DNA. *RTG* (an acronym for ReTroGrade) genes are the major positive regulators of mitochondrial retrograde signaling, with *RTG2* acting as the main upstream regulator and *RTG1* and *RTG3* forming a heterodimer binding the promoters of *RTG*-target genes [1]. *RTG*-dependent retrograde signaling elicits a transcriptional reprogramming of carbon metabolism to re-establish metabolic homeostasis altered by dysfunctional mitochondria [1]. Up-regulation of the peroxisomal isoform of citrate synthase (*CIT2*) serves to replenish the tricarboxylic acid cycle (TCA) cycle and is considered the hallmark of *RTG* pathway activation [2,3]. 

The contribution of mitochondrial retrograde signaling to yeast stress response and cellular adaptation has been described in several cases, including acid stress, endoplasmic reticulum stress, as well as oxidative and osmotic stress [4,5,6,7,8,9]. Particularly noteworthy is the report of the modulation of Rtg1/3 complex activity by the master regulator of osmostress response, the *HOG1* stress activated protein kinase *(SAPK)*, which points to a direct connection between *RTG* genes and targets and the high osmolarity glycerol (*HOG*) signaling system, functionally conserved from yeast to humans [8,10]. In this context, it is also of note that a prominent role of mitochondrial function as an inducible determinant of osmoadaptation has been described [11]. 

Although the picture of yeast osmoregulation is both extensive and detailed, integrated by computational simulations providing an overall holistic understanding, the interplay between and within stress pathways is still a matter of investigation [12,13,14,15,16]. In this regard, the way in which *RTG* signaling contributes to osmoadaptation remains to be elucidated. 

This work aims to provide insight into the role of mitochondrial *RTG* signaling under conditions of salt-induced osmotic stress and its interaction with *HOG1*. Our results support a positive interaction between the two pathways, suggesting a temporal shift in the stress signaling cascade, with the *RTG* pathway acting downstream of *HOG1*. Our data also demonstrate the relevance of the *RTG* pathway in sustaining mitochondrial function under stress, reinforcing the strict relationship between stress response and metabolism in cellular adaptation. 

## 2. Materials and Methods

### 2.1. Yeast Strains and Growth Conditions

The *S. cerevisiae* strains used in this study were W303-1B (WT) cells (MATα ade2 leu2 his3 trp1 ura3) and derivatives *Δ*rtg2 (rtg2*Δ*::LEU2), *Δ*rtg3 (rtg3::LEU2), *Δ*hog1 (hog1*Δ*::NAT#2), kindly provided by Prof. Posas, Universitat Pompeu Fabra, Barcelona, Spain, as well as *Δ*rtg2*Δ*hog1 (rtg2*Δ*::LEU2 hog1*Δ*::NAT#2) [4,5]. Cells were grown at 30 °C in YPD (1% yeast extract, 2% bactopeptone and 2% glucose with 2% agar for solid medium) in the absence or in the presence of sodium chloride (NaCl). Cell growth was monitored qualitatively on YPD agar plates and quantitatively by measuring optical density (600 nm) on liquid YPD medium cultures grown either in micro-well plates or in flasks. 

### 2.2. Micro- and Batch-Culture Growth Assays

For micro- and batch culture-growth assays, fresh overnight pre-cultures were diluted in triplicate in multiwell plates or flasks to the same initial OD_600_. Optical density was then constantly monitored using a high-precision TECAN microplate reader equipped with a shaker and a temperature control unit or by using a Thermo Spectronic Genesis 20 spectrophotometer at selected times. Micro-culture growth curves were analyzed in Microsoft Excel, and the cell growth parameters were determined as follows: the specific growth rate was calculated by plotting optical density as a function of time on a semi-logarithmic scale and extracting the angular coefficient of the obtained equation; doubling time was calculated as reported in [17]; growth efficiency was calculated as the percentage of the maximal cell density reached under stress conditions (with NaCl) compared to the control (without NaCl) [18]. Similarly, relative growth was calculated in batch cultures as a percentage of the optical density values under stress conditions compared to the control at different times. At least three independent cultures were analyzed for each condition in each independent experiment.

### 2.3. Spotting Assay

Fresh overnight yeast cultures (30 °C, YPD medium) were adjusted to the same optical density (OD_600_ = 1), and serial dilutions were spotted on YPD agar medium with or without NaCl. Plates were incubated at 30 °C for 2–5 days, and images were acquired by means of a ChemiDoc Touch Imaging System and analyzed using Image Lab software. 

### 2.4. Quantitative PCR (qPCR) 

The mRNA levels of peroxisomal citrate synthase-encoding gene (*CIT2*) were determined in continuously growing cells after 5 h of NaCl exposure and in the absence of stress. Then, 5 × 10^7^ cells were collected and centrifuged at 3000× *g*. Cell pellets were stored at −80 °C before total RNA extraction with a Presto Mini RNA Yeast Kit (Geneaid, New Taipei City, Taiwan). We immediately performed 0.5 μg RNA (OD_260_/OD_280_ ≥ 2.0) reverse transcription using a QuantiTect Reverse Transcription Kit (Qiagen, Hilden, Germany), and cDNA was directly used for quantitative PCR (qPCR) analysis or stored at −20 °C. The actin 1 (*ACT1*) mRNA was amplified in parallel and used as a housekeeping gene.

Quantitative PCR was carried out on a ABI Prism 7900 HT system from Applied Biosystems using the following primer pairs based on the cDNA sequences of the investigated genes and designed with Primer Express 3.0 (Applied Biosystems, Thermo Fisher Scientific, Waltham, Massachusetts, Stati Uniti). The primer sequences used were: *CIT2*-Forward 5′-TGTAAGGCAATTCGTTAAAGAGCAT-3′ and *CIT2*-Reverse 5′-CCCATACGCTCCCTGGAATAC-3′; *ACT1*-Forward 5′-ACTTTCAACGTTCCAGCCTTCT-3′ and *ACT1*-Reverse 5′-ACACCATCACCGGAATCCAA-3′. The primers were purchased from Invitrogen (Life Technologies). Twenty microliters of reaction volume contained 20 ng of reverse-transcribed first-strand cDNA, 10 μL of SYBR Select Master Mix (Applied Biosystems, Thermo Fisher Scientific, Waltham, Massachusetts, Stati Uniti) and 300 nM of each primer. The specificity of the PCR amplification was checked with the heat dissociation protocol after the final cycle of PCR. The amount of *CIT2* mRNA was normalized with *ACT1* mRNA and calculated in relative units (2^−ΔCt^), where ΔCt is the Ct_sample_−Ct_reference_ gene and Ct is the threshold cycle [19].

### 2.5. Oxygen Consumption Measurements

Respiration was measured in intact cells at 30 °C using an Oxygraph-2 k system (Oroboros, Innsbruck, Austria) equipped with two chambers, and the data were analyzed using DatLab software as described in [20]. Briefly, yeast cells were harvested after 24 h of growth in YPD in the presence or in the absence of NaCl, centrifuged at 3000× *g* for 5 min at 4 °C and resuspended in the same medium to a final optical density of 5 OD_600_ units/mL. Fifty microliters of this suspension, corresponding to about 5 × 10^6^ cells/mL, was added to each chamber containing 2 mL of YPD. The chambers were then closed and respiration recorded.

### 2.6. Statistical Analysis

All the experiments were repeated at least three times, and the results are reported as means with standard deviation. For the determination of significant differences between samples, all results were analyzed using Student’s *t*-test run on EXCEL software, with significant differences indicated at *p* values ** ≤ 0.01 and * 0.05.

## 3. Results

### 3.1. Cell Sensitivity to Osmotic Stress Due to NaCl Treatment 

In a first series of experiments, we analyzed cell sensitivity to osmotic stress by testing the effect of increasing concentrations of NaCl (0.4–1.2 M) on cell growth either in solid rich medium or in continuously growing batch cultures. As reported in Figure 1A, wild-type cells grown in the presence of 0.4 M NaCl did not show significant differences compared to control untreated cells. A relative growth of about 90% was measured by comparing optical density of control and treated cells (0.4 M NaCl) at 24, 48 and 72 h (Figure 1B). As expected, cell sensitivity to osmotic stress increased at higher NaCl concentrations. While 1.2 M was highly toxic in both solid and liquid medium (relative growth of about 30%), exposure to 0.8 M showed an intermediate phenotype on solid medium and a relative growth of about 60% (Figure 1). These data show that cell growth inhibition in the presence of NaCl is dose dependent either on solid or in liquid medium. The concentration of 0.8 M NaCl was selected for the next experiments.

### 3.2. Impairment of RTG Signaling Sensitizes Cells to High Osmostress in an HOG1-Dependent Manner

In order to gain insight into the role of *RTG* signaling and its interaction with the *HOG1 MAPK* pathway under osmostress, we analyzed the effect of *RTG2* and/or *HOG1* deletion on cell growth in the absence and in the presence of 0.8 M NaCl. While no growth differences could be observed across all untreated cells, an evident growth impairment compared to wild-type cells in the absence of the main sensor of the mitochondrial retrograde pathway was revealed under stress (Figure 2). Notably, smaller colonies could be observed in *∆RTG2* cells grown in the presence of NaCl. As expected, the lack of *HOG1* completely abolished growth in a NaCl medium. The double mutant, lacking both *HOG1* and *RTG2*, was unable to grow in the presence of NaCl similarly to the single *HOG1* mutant. 

To further examine cell growth and the physiological features of the osmostress response, we performed batch culture growth assays of wild-type and mutant cells in the presence and in the absence of NaCl. In fact, growth curve assays in liquid cultures are highly sensitive and able to uncover the subtle phenotypes as compared with cell dilution spot tests on solid media. Cell growth was monitored up to 50 h, and curves from untreated cells were compared to curves from treated cells together with the estimation of kinetic growth parameters (Figure 3 and Table 1). In all cases, growth inhibition due to NaCl treatment could be observed. Specific growth rate was reduced by about 50%, and doubling time increased from 2.0 to 3.9 h in wild-type cells under stress (Figure 3a and Table 1). Deletion of *RTG2* exacerbated this phenotype by showing a doubling time of 5.9 h in a NaCl medium with an increase of 1.5-fold compared to wild-type cells. A significant decrease in exponential growth rate was also measured in *RTG2*-lacking mutants compared to wild-type cells—0.17 and 0.12 h^−1^, respectively, under NaCl treatment (Figure 3b and Table 1). According to the literature, the highest doubling time was measured for *∆HOG1* cells in NaCl (10.3 h) with the lowest exponential growth rate, 0.08 h^−1^ (Figure 3c and Table 1). Notably, a slight growth improvement compared to *∆HOG1* was observed in *∆HOG1∆RTG2,* which showed a doubling time of 7.2 h with an exponential growth rate of 0.12 h^−1^ (Figure 3d and Table 1). All untreated wild-type and mutant cells show comparable growth features (Figure 3 and Table 1). Growth efficiency calculated in all samples as the percentage of the maximal cell density reached under stress conditions (with NaCl) compared to the control (without NaCl) mirrored these results, highlighting significant differences among the mutants compared to wild-type cells (Table 1). 

Overall, these data demonstrate the involvement of *RTG2* in osmoadaptation, confirm that *HOG1* is essential for cell survival in the presence of high salt stress and support the interaction between *HOG1* and *RTG* signaling under hyperosmotic conditions. 

To further characterize the involvement of *RTG2* in osmoadaptation, we performed growth assays by using batch cultures in shaking flasks and analyzed cell proliferation at different times in wild-type and mutant cells. Relative cell growth was measured at 8, 24, 48 and 72 h corresponding to different growth phases referred to as the exponential phase, the diauxic shift and the stationary phase, respectively. As reported in Figure 4, relative growth in the presence of NaCl was similar in wild-type and *∆RTG2* exponential phase cells (about 20%) but significantly different with respect to *∆HOG1* and *∆HOG1∆RTG2* (about 8%) after 8 h. However, a highly significant difference could be observed comparing wild-type and *∆RTG2* relative growth at 24 h, about 60% versus 40%. In the same samples, less significant differences in relative growth could be observed at 48 and 72 h. Relative growth was below 10% in *∆HOG1* and *∆HOG1∆RTG2* at all times analyzed. These results indicate that *RTG2* is especially required for osmoadaptation after 24 h of growth (corresponding to 4–5 generations)—that is, when glucose becomes limiting and cells are characterized by a decreased growth rate and a metabolic switch towards respiration. 

To verify whether *RTG2* acts through *RTG* signaling in osmoadaptation, we analyzed cell growth of *∆RTG3* cells in solid and liquid rich medium with or without NaCl. As reported in Figure 2, growth impairment could be observed in *∆RTG3* cells with respect to the wild type and with similar features of *∆RTG2.* Growth curves obtained for *∆RTG3* indicate a 1.4-fold increase in NaCl doubling time (5, 4 h) with respect to wild-type cells, with a specific growth rate and growth efficiency of 0.12 h^−1^ and 80%, respectively, similar to the results for *∆RTG2* (Figure 3 and Table 1). As for *∆RTG2,* a significant difference in relative growth compared to the wild type was observed after 24 h in shaking batch cultures (Figure 4). 

These data indicate that *RTG2* contributes to *HOG1-*dependent osmoadaptation via *RTG* signaling and confirm previous results on the interplay between *RTG3* and *HOG1* under stress [8].

### 3.3. RTG Signaling Is Activated in Osmoadaptation 

Since *CIT2* up-regulation is considered a hallmark of *RTG*-dependent retrograde signaling activation [1,3], and having observed growth impairment of *RTG* mutants, especially in a late growth phase, its expression was evaluated after 5 h of growth (judged as a time of long-term transcriptional response [18]) in the presence and in the absence of NaCl stress. As shown in Figure 5, NaCl treatment causes a significant increase in *CIT2* mRNA expression level (about 7-fold), which is considerably reduced in the absence of *RTG2*, where a reduction of about 8-fold is observed (Figure 5). As expected, *CIT2* was down-regulated in *RTG2*-lacking cells compared to the wild type under both control and stress conditions, showing that *RTG*-dependent mitochondrial retrograde signaling is activated as a result of long-term adaptation to osmostress. 

### 3.4. RTG Mutants Show Decreased Respiratory Capacity under Osmostress 

Having observed a major involvement of the *RTG* pathway in a late phase of stress response in cell growth features and in signaling activation, we hypothesized its major requirement during the transition from fermentation to respiration, known to be characterized by structural and functional reorganizations in mitochondrial metabolism [21]. To address this point, mitochondrial respiratory capacity was evaluated in whole cells in the presence and in the absence of NaCl after 24 h of growth. Wild-type cells showed a similar oxygen consumption rate with or without NaCl (Figure 6). In both untreated *RTG* mutants, respiratory capacity was slightly improved compared to wild-type cells but was significantly reduced in the presence of NaCl, showing a decrease of about 2.5-fold in both untreated mutants and wild-type treated cells (Figure 6). 

These data demonstrate that osmoadaptation at the level of mitochondrial function is successfully completed within 24 h of growth, but impairment of *RTG* signaling negatively affects respiratory capacity in growing cells upon high osmostress. 

## 4. Discussion

This work highlights the relevance of the interplay between signaling and metabolism in cellular stress response. We demonstrated that *RTG*-dependent retrograde signaling sustains mitochondrial function in a late phase of yeast osmoadaptation still dependently on *HOG1*, the major regulator of HOG signaling. The role of *RTG* signaling and its interaction with *HOG1* were investigated in wild-type and mutant cells on solid and liquid mediums, i.e., batch cultures, under high osmotic stress conditions induced by NaCl treatment.

We show that the lack of either *RTG2*, the main sensor of *RTG* signaling, or the transcriptional factor *RTG3* interferes with successful osmoadaptation in a manner dependent on *HOG1* (Figure 1 and Figure 2). These data demonstrate that *RTG* signaling contributes to osmostress resistance and confirm that *HOG1* plays a role upstream of *RTG2* in the stress signaling cascade, further supporting the interaction between these two pathways under cellular stress [5,8]. This is in agreement with previous work indicating that *RTG* signaling plays a protective role in stress resistance [7]. 

Growth measurements of batch cultures in the presence or absence of osmostress gave us the opportunity to compare the real-time growth and physiological features of wild-type and mutant cells [12]. 

The growth differences among wild-type and mutant cells confirmed the *HOG1*-dependence of osmoadaptation even in the absence of *RTG2* (Figure 3 and Figure 4 and Table 1). The concomitant abolition of *HOG1* and *RTG2* results in a slight improvement in cell growth in the presence of NaCl, which seems to be evidence of the occurrence of a masking interaction within the sub-class of asymmetric and positive genetic interactions occurring between *HOG1* and RTG2, as reported in [22]. Our results also show that *RTG* signaling could maintain mitochondrial functionality only in response to stress and not during normal growth. However, we showed that *RTG* pathway activation together with the de-repression of respiratory capacity induces acetic-acid stress resistance [4]. Thus, we cannot exclude the involvement of other osmostress regulators whose activity might be negatively affected by the *HOG-RTG* signaling axis. 

Data deriving from the determination of relative growth in the presence of salt stress highlighted that *RTG* signaling is especially required in a late phase of stress response, between 16 and 24 h of growth—that is, after 4–5 generations—presumably corresponding to a decrease in glucose concentration accompanied by a metabolic transition towards respiration (Figure 4). In this context, it is plausible that *RTG* signaling supports the reprogramming of carbon metabolism under osmostress, as observed in [23]. Accordingly, *RTG* signaling activation was found as expected by *CIT2* up-regulation in WT-stressed cells compared to no stress conditions (Figure 5). On the other hand, the very low level of *CIT2* in the stressed *∆RTG2* mutant confirms *RTG* signaling as the major regulator of its expression also under conditions of osmostress. In this regard, considering the fact that Cit2 functions in the glyoxylate cycle, the relevance of peroxisomes in the adaptation of yeast cells to an osmotic environment is reinforced [18]. The low but significant increase in *CIT2* expression under stress in *RTG2*-lacking cells can be explained by additional regulatory mechanisms, including the up-regulation of this gene by Msn2/Msn4 under glucose-limiting conditions to support peroxisomal function [9,24] or through interaction with the carbon catabolite repression pathway [25].

Our data also show that mitochondrial function is required for successful osmoadaptation, as determined by the comparable respiratory activity in the presence and in the absence of stress conditions (Figure 6). This, together with decreased basal respiration in *RTG* mutants exclusively under stress, provides direct evidence that *RTG* signaling sustains mitochondrial respiratory function in a high osmotic environment (Figure 6). Thus, both RTG signaling activation and the maintenance of mitochondrial function are confirmed as powerful adaptive strategies under conditions of stress [26,27].

These results provide the first evidence that *RTG* signaling contributes to ensuring mitochondrial function as part of a *HOG1*-mediated osmoadaptive response, as has previously been suggested [8]. Interestingly, impairment of *RTG* signaling improves mitochondrial respiration in the absence of stress conditions, suggesting an interaction between this pathway and other regulators of mitochondrial activity, as already observed in [2]. 

Mitochondria are important determinants in osmoadaptation, and, here, we show the contribution of *RTG* signaling to the significant functional changes occurring at a mitochondrial level as a response to glucose depletion [11,21,28]. 

The temporal and dynamic interactions between *HOG1* and the *RTG* pathway under hyperosmotic conditions are a matter of speculation. *HOG1* is known to be required in the well understood first line of cellular defense against stress (the rapid-and-transient mode). In this phase, glucose is diverted from glycolysis to glycerol biosynthesis in order to rapidly support the cellular stress response program [10,29]. 

*RTG* signaling might be involved in a later stage of adaptation related to metabolic adjustment for respiration (Figure 7). This is in agreement with the separateness (but note the interdependence) of the time scales between rapid signaling events and the relatively slower downstream adaptive processes in response to hyperosmolarity, which also affects long-term memory in response to periodic stress conditions [30]. 

The down-regulation of the peroxisomal isoform of citrate synthase and the impaired mitochondrial respiration observed in the *RTG* mutants suggest the importance of inter-organellar crosstalk for maintenance of cellular homeostasis under changing environmental conditions. In particular, the metabolic communication between peroxisomes and mitochondria in long-term adaptation to salt stress appears to be confirmed [9,18]. In this respect, *RTG* signaling could provide citrate via the glyoxylate cycle to sustain mitochondrial function under stress conditions. 

Overall, our results indicate the sequential cooperation between the two stress signaling pathways *HOG* and *RTG* to accomplish cellular adaptation and the central role of *RTG* signaling in maintaining mitochondrial function under stress conditions. Further research to identify the mechanisms involved in the activation of the *RTG* pathway in osmotic and other stress responses will help to elucidate the temporal cascade in the stress signaling network and its connections with metabolic clues. 

## Figures and Tables

**Figure 1 microorganisms-09-01894-f001:**
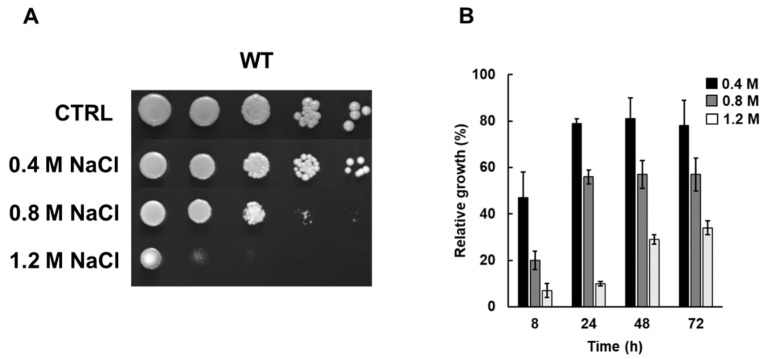
Sensitivity of wild-type cells to sodium chloride. Wild-type cells (WT), grown overnight in YPD medium, were: (**A**) diluted to 1 OD_600_, and ten-fold serial dilutions were spotted on YPD plates with or without sodium chloride (NaCl) at the indicated concentrations, with growth being scored after 2–5 days; (**B**) diluted to 0.1 OD_600_ in fresh liquid YPD with or without NaCl at the indicated concentrations, and optical density (OD_600_) was measured at the indicated times. Relative growth was calculated as the percentage of the OD_600_ of stressed/control cells.

**Figure 2 microorganisms-09-01894-f002:**
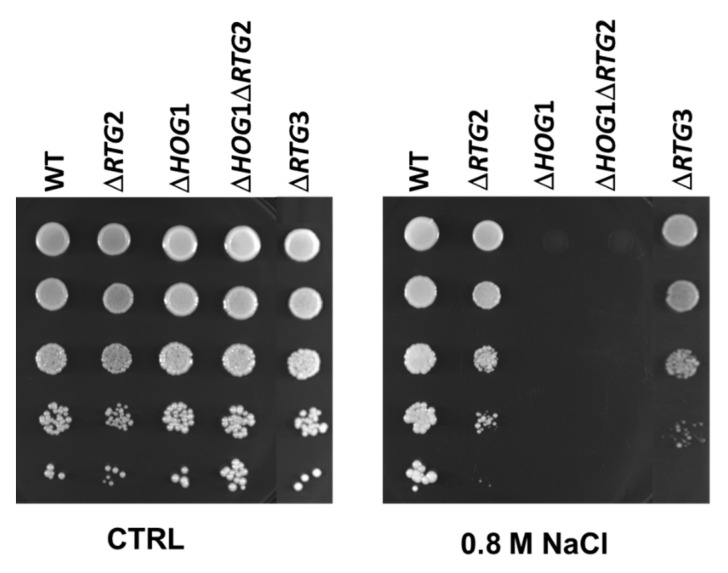
Sensitivity of wild type and mutant cells to sodium chloride. Wild-type (WT) and mutant cells, lacking *RTG2 (**∆RTG2**), RTG3 (**∆RTG3)* and/or *HOG1 (**∆HOG1;*
*∆HOG1**∆RTG2)*, were grown overnight in YPD medium and diluted to 1 OD_600_, and ten-fold serial dilutions were spotted on YPD plates without (CTRL) or with 0.8 M sodium chloride (NaCl). Growth was scored after 2–5 days.

**Figure 3 microorganisms-09-01894-f003:**
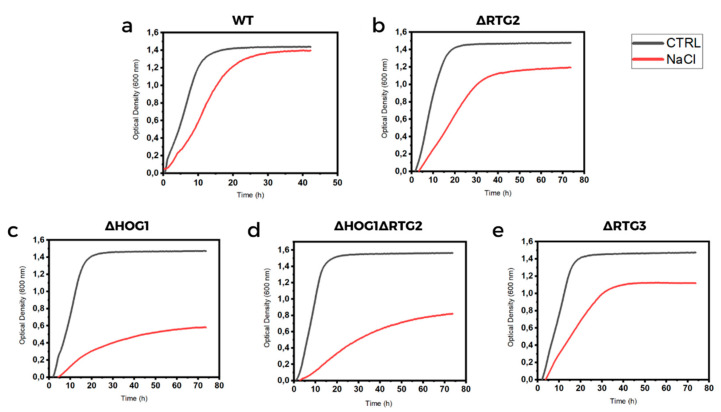
Micro-culture growth curves of wild-type and mutant cells. (**a**) Wild-type (WT) and (**b**–**e**) indicated mutant cells (*∆RTG2*, *∆HOG1*, *∆HOG1**∆RTG2* and *∆RTG3*) grown overnight in YPD medium were diluted to 0.01 OD_600_ in fresh liquid YPD with or without 0.8 M sodium chloride (NaCl), and optical density was measured at 600 nm (OD_600_) over time with a high-precision TECAN microplate reader. Each experiment was performed in triplicate, and representative micro-culture growth curves from three independent experiments are reported.

**Figure 4 microorganisms-09-01894-f004:**
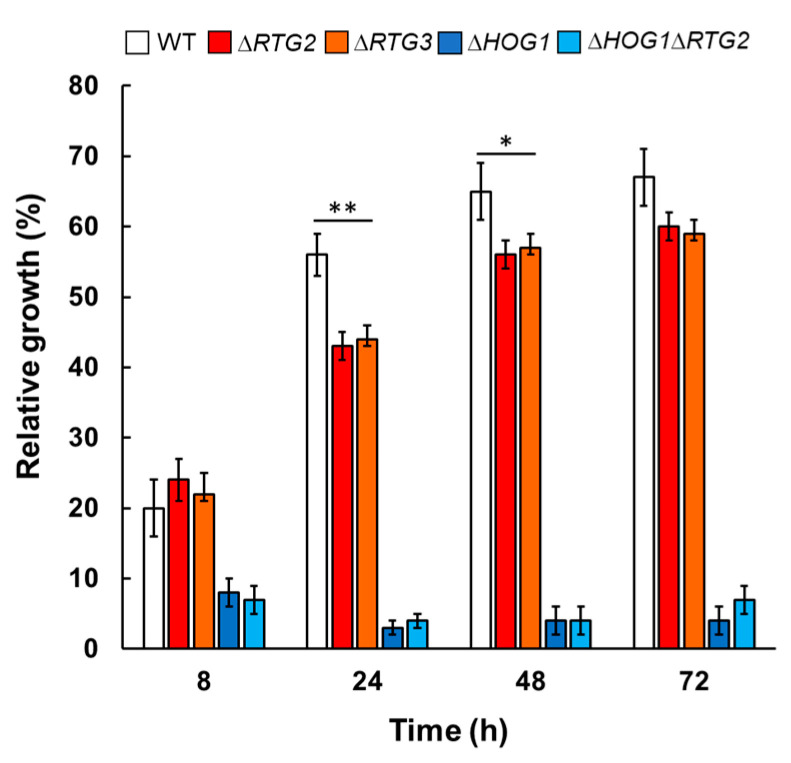
Relative growth of wild-type and mutant cells in batch cultures in the presence of sodium chloride. Wild-type (WT) and mutant cells (*∆RTG2*, *∆HOG1*, *∆HOG1**∆RTG2* and *∆RTG3*), grown overnight in YPD medium, were diluted to 0.1 OD_600_ in fresh liquid YPD with or without 0.8 M sodium chloride (NaCl), and optical density (OD_600_) was measured at 600 nm at the indicated times. Relative growth was calculated as the percentage of the OD_600_ of stressed/control cells. Unpaired Student’s *t*-test: statistical significance differences with ** *p* < 0.01 and * *p* < 0.05 when comparing wild type with *RTG* mutants from six independent experiments at 24 h.

**Figure 5 microorganisms-09-01894-f005:**
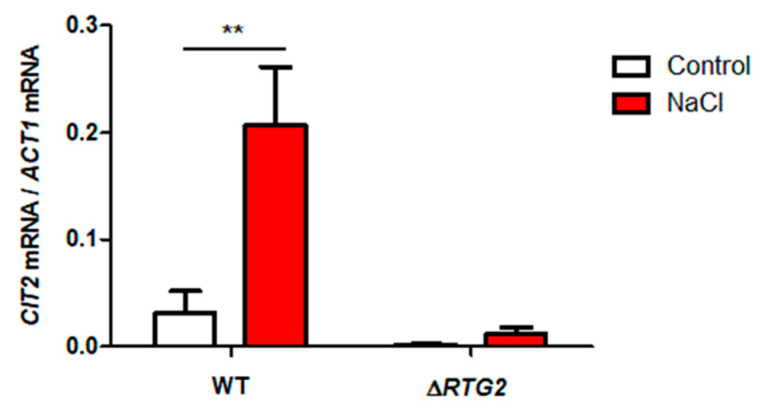
*CIT2* expression under high osmotic environment. Wild-type (WT) and *∆RTG2* cells, grown overnight in YPD medium, were diluted to 0.1 OD_600_ in fresh liquid YPD with or without 0.8 M sodium chloride (NaCl). After 5 h, cells were collected for RNA extraction, and mRNA levels were measured by quantitative PCR. The amount of *CIT2* mRNA was normalized with *ACT1* mRNA and calculated in relative units (2^−ΔCt^), where ΔCt is the Ct_sample_−Ct_reference_ gene and Ct is the threshold cycle. Unpaired Student’s *t*-test: a statistically significantly difference with ** *p* < 0.01 when comparing wild-type untreated cells versus NaCl-stressed cells from three independent experiments.

**Figure 6 microorganisms-09-01894-f006:**
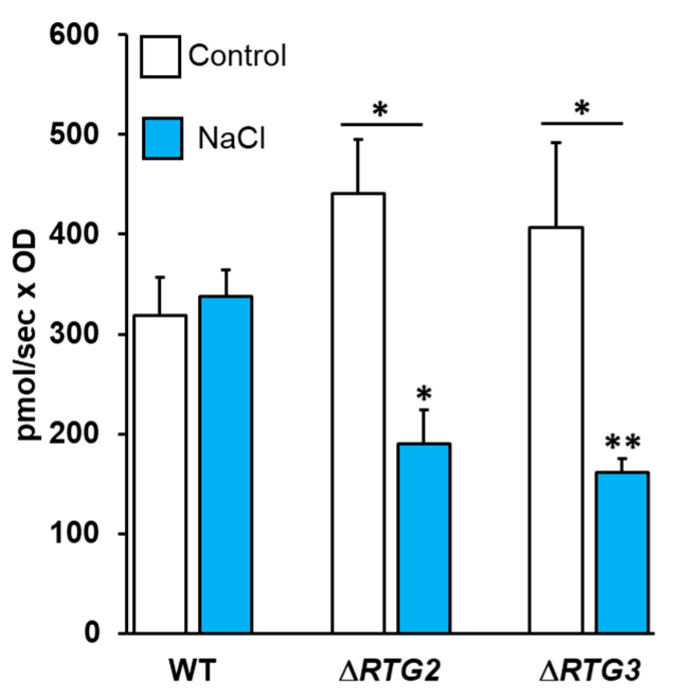
Mitochondrial basal respiration of wild-type and *RTG* mutants. Wild-type (WT), *∆RTG2* and *∆RTG3* cells, grown overnight in YPD medium, were diluted to 0.1 OD_600_ in fresh liquid YPD with or without 0.8 M sodium chloride (NaCl). After 24 h, cells were collected and resuspended in the same medium to a final optical density of 5 OD_600_ units/mL. Oxygen consumption was measured with an Oxygraph-2 k system, and experimental data were analyzed using DatLab software. Unpaired Student’s *t*-test: a statistically significantly difference with * *p* < 0.05 when comparing *∆RTG2* and *∆RTG3* control versus NaCl, respectively, or wild-type NaCl versus *∆RTG2* NaCl, and ** *p* < 0.01 when comparing wild-type NaCl versus *∆RTG3* NaCl from three independent experiments.

**Figure 7 microorganisms-09-01894-f007:**
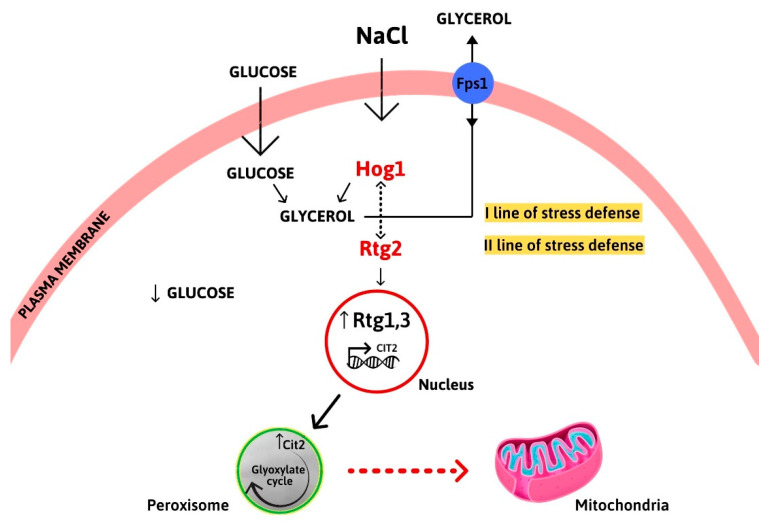
Interplay between *HOG1* and the *RTG* pathway in osmoadaptation. Hog1 is activated by osmostress to promote glycerol biosynthesis as a rapid and transient cellular stress response. Intracellular glycerol accumulation is controlled by its own synthesis and by the aquaglyceroporin Fps1p. *RTG* signaling is activated in a later phase, as determined by *CIT2* up-regulation, to sustain mitochondrial function and metabolic homeostasis for cellular adaptation.

**Table 1 microorganisms-09-01894-t001:** Growth parameters of reported yeast strains in the absence and in the presence of sodium chloride.

Yeast Strains	Specific Growth Rate (µ_max_ h^−1^)	Doubling Time (h)	Growth Efficiency (%)
WT	0.38 ± 0.02	2.0 ± 0.32	-
WT + NaCl	0.17 ± 0.01	3.9 ± 0.71	87 ± 2
*∆* *RTG2*	0.39 ± 0.03	2.2 ± 0.07	-
*∆**RTG2* + NaCl	0.12 ± 0.01	5.9 ± 0.12	80 ± 4
*∆* *HOG1*	0.32 ± 0.01	2.1 ± 0.4	-
*∆**HOG1* + NaCl	0.08 ± 0.02	10.3 ± 0.2	44 ± 9
*∆* *HOG1* *∆* *RTG2*	0.29 ± 0.09	2.5 ± 0.78	-
*∆**HOG1**∆**RTG2* + NaCl	0.12 ± 0.01	7.2 ± 0.2	56 ± 12
*∆* *RTG3*	0.37 ± 0.04	2.5 ± 0.14	-
*∆**RTG3* + NaCl	0.12 ± 0.01	5.4 ± 0.71	80 ± 2

Growth rate, doubling time and growth efficiency of wild-type (WT) and mutant cells (*∆RTG2*, *∆HOG1*, *∆HOG1**∆RTG2* and *∆RTG3*) grown in the absence and in the presence of sodium chloride (NaCl) are the mean ± standard deviation of three independent experiments, each performed in triplicate.

## Data Availability

Raw data from this study are available on reasonable request from the corresponding author.

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
