# Peer review of "RTG Signaling Sustains Mitochondrial Respiratory Capacity in HOG1-Dependent Osmoadaptation"

_microorganisms, 2021, doi:10.3390/microorganisms9091894_

Round 1

Reviewer 1 Report

The article nicely establishes the background, confirms that NaCl has a dose-dependent negative effect on yeast growth and shows that deleting HOG1 abolishes growth at high NaCl while deletion of retrograde signal mediator genes RTG2 or RTG3 reduce growth. It goes on to show that the growth reduction is highly significant at 24 hours and significant at 48 hours. NaCl induces higher expression of RTG target gene CIT2 in a manner that is dependent on RTG2. Finally, it is shown that NaCl causes a >50 % reduction in respiration but only in the absence of RTG2 or RTG3, not in the wild type.

The article is well set out and easy to follow. The methods are clear and concise and the results (including the figures and table) support the conclusions drawn in the discussion. The findings are placed in the context of relevant previous reports and speculation regarding their implications is particularly well argued. The comments regarding peroxisomal involvement for example.

This article will appeal to a wide readership and builds upon previous ground-breaking work by the team in the area of stress responses. The results and conclusions are sound and highly suitable for publication in the journal.

The only other comment I would make is that some minor English language changes should be made:

47/48: "it is of also of note a prominent role of mitochondrial function as"  should be something like " it is also of note that a prominent role of mitochondrial function is as"

51: "at" should be "in"

52: "is far to be elucidated" could be "remains to be elucidated" or "is far from understood"

77: "logarithmic" spelt "logaritmic"

95: OD wavelengths should be lower case

118: OD600nm: delete "nm"

Figures 1 and 5 use a mixture of "." and "," for decimal points

Figures 1 and 2: "serially dilutions series" should be "serial dilution series" or "serial dilutions"

154: mutant names in upper case instead of lower

161: delete "the"

Figure 3: mutant names in upper case

Table I: mutant names suggest proteins rather than genes

201: "referred to" should be "referred to as"

Figure 4: lacks label on x axis

226/227: should one "RTG2" be changed to "RTG3"?

295: "the evidence"  should be "evidence"

326: "give first evidence" should be "provide the first evidence"

Reviewer 2 Report

The authors evaluated the role of RTG signaling in salt-induced osmotic stress and its interaction with HOG1. They compared in wild type and mutants (lacking HOG1 and/or RTG genes) cell growth features, retrograde signaling activation and mitochondrial function in the presence and in the absence of high osmostress.

They described in detail the S. cerevisiae growth conditions and they used several experimental approaches including spotting assay, qPCR and oxygen consumption (using high-resolution respirometry). The experiments were well designed and performed. The results are clearly presented.

The data presented support the suggestion that HOG1 and RTG may interact sequentially in the stress-signaling cascade.

Major points

-The authors must be specific in the methodology of how they obtained the mutant cells.

-The authors should write a conclusion of their results.

Minor points

-It is suggested to avoid abbreviations in the title.

-All the abbreviations must be defined the first time used.

-All the abbreviations used in the figures must be defined in the figure legend. Figures should be understood without reading the text.

-Table 1. The title of the table should be above the table and the description of the table should be under.

-A space should be inserted between lines 197 and 198 (to separate the table from the text)

Reviewer 3 Report

The article by Guaragnella et al, describes an interesting mechanism as to how mitochondrial retrograde gene signaling cordinates with HOG signaling during osmoadaptation. The article is well written, however certain experiments are needed to be carried out to better support the inference of the article. Here are my comments:

1) One important aspect of mitochondrial retrograde signaling is analyzing the changes in mitochondrial membrane potential (MMP) as previously described in https://pubmed.ncbi.nlm.nih.gov/26583058/

The authors should measure the MMP in the wild-type and the mutants under osmotic stress following simple experiments as described previously (https://pubmed.ncbi.nlm.nih.gov/22303396/)

2)Further osmotic stress adaptation via HOG pathway is dependant on the ergosterol biosynthesis as discussed previously (https://www.ncbi.nlm.nih.gov/pmc/articles/PMC7397035/), several altered expression of ergosterol biosynthesis genes that include ERG10,ERG13,HMG1/2,ERG19,ERG20,ERG9, and so on (described in https://pubmed.ncbi.nlm.nih.gov/30042199/) showed altered growth rates under osmotic stress. The authors should check the expression of the above mentioned genes in the mutants in presence of osmotic stress with qPCR, since HOG pathway regulates ergosterol synthesis. Also altered expression of ergosterol biosynthesis genes affect cellular growth in glycerol containing media. This will make the article stronger.

3)The authors only analyzed mitochondrial respiration, whereas ATP levels and mitochondrial ROS can affect the mitochondrial retrograde signaling as described before https://www.ncbi.nlm.nih.gov/pmc/articles/PMC3266616/ 

The authors should consider analyzing mitochondrial ROS and ATP levels using simple experiments as described previously (https://pubmed.ncbi.nlm.nih.gov/34207384/) 

4) Further, CIT2 expression should also be measured in the other mutants apart from RTG2 mutant in presence of 0.8 M NaCl.

5) Fig 6 the basal mitochondrial respiration should also be checked in hog1 mutant along with hog1rtg2 double mutant

Round 2

Reviewer 3 Report

The authors have answered satisfactorily to all my queries.